# Self-Association of the Anion of 7-Oxodeoxycholic Acid (Bile Salt): How Secondary Micelles Are Formed

**DOI:** 10.3390/ijms241411853

**Published:** 2023-07-24

**Authors:** Mihalj Poša

**Affiliations:** Department of Pharmacy, Faculty of Medicine, University of Novi Sad, Hajduk Veljka 3, 21000 Novi Sad, Serbia; mihaljp@uns.ac.rs

**Keywords:** micelles, bile acids, Le Chatelier’s principle, critical micellar concentration

## Abstract

Bile acid anions are steroidal biosurfactants that form primary micelles due to the hydrophobic effect. At higher concentrations of some bile acid anions, secondary micelles are formed; hydrogen bonds connect primary micelles. Monoketo derivatives of cholic acid, which have reduced membrane toxicity, are important for biopharmaceutical examinations. The main goal is to explain why the processes of formation of primary and secondary micelles are separated from each other, i.e., why secondary micelles do not form parallel to primary micelles. The association of the anion of 7-oxodeoxycholic acid (a monoketo derivative of cholic acid) is observed through the dependence of the spin–lattice relaxation time on total surfactant concentration *T*_1_ = *f*(*C_T_*). On the function *T*_1_ = *f*(*C_T_*), two sharp jumps of the spin–lattice relaxation time are obtained, i.e., two critical micellar concentrations (CMC). The aggregation number of the micelle at 50 mM total concentration of 7-oxodeoxycholic acid anions in the aqueous solution is 4.2 ± 0.3, while at the total concentration of 100 mM the aggregation number is 9.0 ± 0.9. The aggregation number of the micelle changes abruptly in the concentration interval of 80–90 mM (the aggregation number determined using fluorescence measurements). By applying Le Chatelier’s principle, the new mechanism of formation of secondary micelles is given, and the decoupling of the process of formation of primary and secondary micelles at lower concentrations of monomers (around the first critical micellar concentration) and the coupling of the same processes at higher equilibrium concentrations of monomers (around the second critical micellar concentration) is explained. Stereochemically and thermodynamically, a direct mutual association of primary micelles is less likely, but monomeric units are more likely to be attached to primary micelles, i.e., 7-oxodeoxycholic acid anions.

## 1. Introduction

According to Small, bile salts are biologically the most important detergent-like molecules [1]. Bile acid anions, similar to classic surfactants with a hydrophobic hydrocarbon chain (tail) and a polar head, in terms of hydrophobicity, have two regions (Figure 1): a more hydrophobic convex surface (β-side of the steroid skeleton) and a less hydrophobic (i.e., hydrophilic) concave surface (α-side of the steroid skeleton) [1,2,3,4,5,6,7,8,9,10] (Appendix A). Regarding bile acids, cholic acid is the most complete separation of the hydrophobic and hydrophilic surfaces. Namely, C7 and C12 OH groups are α-axial, while the C3 OH group has an α-pseudoaxial configuration [11]. From this, all three OH groups are localized on the concave surface of the steroid skeleton—planar polarity (Figure 1) [3,4,10,11,12,13,14]. By changing the orientation of the OH group from α-axial to α-equatorial or β-equatorial, the hydrophobicity of the convex surface of the steroid skeleton is reduced, which is reflected in the property of bile acid anions, for example, in an aqueous solution, to have a tendency towards a reduction in the self-association of bile salts. It is similar when an α-axial OH group is oxidized to an oxo group—the hydrophobic surface on the β-side of the steroid skeleton decreases [4,5,7,11] (Appendix A).

The anions of bile acids (bile salts) solubilize the insoluble components of bile, such as lecithin and cholesterol, and aid digestion by removing the products of pancreatic hydrolysis (monoglycerides and fatty acids) [1,2,3]. It is well known that some derivatives of bile acid anions have promoting effects on the transport of drugs through the cell membrane, i.e., improving permeability [15,16,17]. Bile acid anions with certain drugs can form mixed micelles [18,19] or molecular aggregates in concentration ranges where they do not form micelles [20,21], changing the bioavailability of drugs [8]. Bile acid anions increase the solubilization of certain drugs, so they have potential applications in pharmaceutical formulations [8,22,23,24]. Salts of hydrophobic bile acids show membranolytic properties (salts of deoxycholic acid). The more hydrophobic a bile acid anion is, the greater its membranolytic activity and solubilization capacity towards the hydrophobic molecular guest [7,25,26,27]. All the properties mentioned above of bile salts result from the self-association properties of bile acid anions, i.e., the tendency to form micelles and reduce the steroid skeleton hydrophobic surface exposure to hydration [1,7].

Small proposed the concept of the formation of primary and secondary micelles [1]. Bile acid anions in primary micelles occupy such a position that their hydrophobic sides are opposite (back-to-back association)—the concept of hydrophobic association. At the same time, the polar groups are oriented towards the water environment (Figure 2) [1,6,9]. Evidence of the formation of primary micelles comes from ^1^H NMR experiments, namely for bile acid salts above the critical micellar concentration (CMC); when primary micelles are formed, there is a broadening of the line (in the NMR spectrum) of C18 and C19 methyl groups from the hydrophobic side of the steroid skeleton [28,29]. About 10 (aggregation number) bile acid anions can be packed into the primary micelles without creating a cavity [1]. Gouin and Zhu concluded that Small’s dimmer micelle plays a crucial role in the self-association mechanism of bile salts. Dimeric micelles can be detected below the CMC value [30]. In Kawamura’s micelle model (which corresponds to Small’s primary micelles with a higher aggregation number ≈ 10), bile acid anions are arranged on the surface of the roller coating so that the angular methyl groups are oriented toward the interior of the geometric body. In contrast, OH groups are oriented toward the aqueous phase. The side chains with carboxylate groups are located on the bases of the roller, where each carboxylate group with the carboxylate group of the adjacent monomer is trans-oriented. In this way, the electrostatic repulsion in the aggregate is reduced [31].

Small applies a phase separation model to describe the thermodynamic process of primary micelle formation, according to which micelle formation is carried out in an all-or-nothing process. Such a model assumes a monodisperse system, sudden changes in the properties of the aqueous solution at the critical micellar concentration and a constant concentration of bile acid anions (monomers) above the CMC.

In the case of secondary micelles, according to Small’s concept, primary micelles at the total concentrations of surfactants significantly above the CMC form hydrogen bonds with each other (also in the all-or-nothing process). For the formation of secondary micelles, Mazer et al. proposed a mechanism of successive association reactions [32]:(1)$${PM}_{n}+{PM}_{n}↔{SM}_{2n},\;{PM}_{n}+{SM}_{2n}↔{SM}_{3n},\;{PM}_{n}+{SM}_{\left(i-1\right)n}↔{SM}_{in},$$
where ${PM}_{n}$ is the primary micelle with aggregation number *n*, while ${SM}_{in}$ corresponds to the secondary micelle consisting of the *i* primary micelle. Molecular dynamic simulations of bile acid salt systems yield aggregates where primary and secondary micelles are recognized [33,34,35,36].

In the vicinity of the CMC, Thomas and Christian rejected Small’s phase separation model (all-or-nothing process) of bile salt association and proposed a stepwise association of primary micelles [37]. They suggested that secondary micellization occurs via hydrogen bonds between hydroxyl and carboxylate groups in an alkaline environment, giving a disc-shaped structure (corresponding to Small’s secondary micelles).

Ekwall, Fontell, and Sten concluded that there are three concentration limits in the self-association process of bile salts [38]: below limit I (sodium cholate: 13 mM–15 mM; sodium deoxycholate: 4 mM–6 mM), there is no micelle formation (according to Gouin and Zhu, in this area, it is possible to form dimeric Small’s primary micelles [30] that correspond to Rusllan’s micellar embryos [39]). Tiny primary micelles are formed between limits I and II, while above limit II (sodium cholate: 45 mM–50 mM; sodium deoxycholate: 9 mM–10 mM), the aggregates are somewhat larger and probably correspond to secondary micelles. Above limit III (sodium cholate: 60 mM–110 mM; sodium deoxycholate: 40 mM–50 mM), the degree of association increases, and aggregates of colloidal dimensions are formed.

Rovnyak et al., based on NMR chemical shift titrations and using the global fitting method for deoxycholic acid anions, found four different critical micellar concentrations: 3.8 mM, 9.1 mM, 27 mM, and 57 mM (298 K, pH 12)—the multi-CMC-phase separation model [29,40]. Matsuoka and Moroi also distinguished several different values of the critical micellar concentration using the fluorescence probe method in bile salts [41,42,43].

The specific geometry of the steroid skeleton—planar polarity—results in bile acid anions forming relatively small primary micelles of 4 to 15 monomer units compared to classical surfactants (while the aggregation number of classical surfactants can be higher than 100) [1,9,44,45]. Micelles of classic ionic surfactants bind relatively large amounts of counter ions, which is not the case with micelles of bile acid anions. Namely, Stern’s boundary layer is formed in micelles of classic surfactants. In contrast, since the carboxylate groups are distant in bile acid anion micelles, Stern’s layer does not form, i.e., the collective action of carboxylate anions is absent [46]. However, classic surfactants and bile acid salts share that at lower temperatures, micelles (primary micelles) are formed through entropic driving forces—the hydrophobic effect. With the increase in temperature, the micellization process of classic surfactants and bile acid salts has an enthalpic driving force [4,13,47,48,49,50,51,52,53,54,55,56,57,58] (Appendix B).

Monoketo (oxo) derivatives of bile acids (salts), as well as the bile acid anion from the title of the manuscript (7-oxodeoxycholic, Figure 3), have a significant application in biomedicine (and in pharmaceutical formulations), since the presence of the keto group reduces the hydrophobic surface on the convex side of the steroid skeleton, which means that their membrane toxicity (membranolytic activity) decreases. However, they still form micelles (their CMC value increases) [7,8,45].

The goal is to continue the preliminary examination of the self-association of 7-oxodeoxycholic acid anions, where the self-association is monitored based on the concentration dependence of the spin–lattice relaxation times for the H atoms of the C18 methyl groups. We aim to provide a new thermodynamic analytical theory explaining secondary micelle formation. Specifically, we want to determine whether these micelles are formed through the linear polymerization of primary micelles or by adding monomers to primary micelles. We will also investigate the relationship between the formation of primary and secondary micelles and how they are mutually coupled or decoupled. Ultimately, we want to identify the thermodynamic theory that explains the functional dependence of the spin–lattice relaxation time of selected protons from the steroid skeleton’s β-side on the total concentration of anions of 7-oxodeoxycholic acid.

In an aqueous solution, it is known that the NaCl promotes the bile salt’s self-association [1,6]. Therefore, the effect of NaCl on the micellization process of 7-oxodeoxycholate is being investigated. The concentration of NaCl in the aqueous solution of the examined bile salt is 50 mM. The concentration of NaCl was chosen for two reasons: First, mainly the dehydration of OH groups (steroid skeleton) is observed at 300 mM NaCl (NaCl promotes the formation of H-bonds, i.e., secondary micelles [1]). However, suppose that the formation of secondary micelles is also affected by a concentration of 50 mM NaCl in an aqueous solution. In addition to H-bonds, the hydrophobic effect is also important. Second, the high salt concentration in NMR samples is challenging because it can prevent precise tuning and matching of the probe head. Also, pulse calibration could be more precise. In extreme cases, damage to the equipment is possible. Thus, keeping salt concentration as low as possible or avoiding using salt altogether is desirable [59].

Knowing the total concentrations of surfactant at which primary and secondary micelles are formed can be important in pharmaceutical formulations. For example, if 7-oxodeoxycholic acid anion micelles are used to solubilize a hydrophobic drug (due to low membranotoxicity), then if H-bonds are present in the secondary micelles (in addition to hydrophobic interactions) and the solution contains urea (which destroys H -connections between micellar building units [1]) the micellar system (i.e., pharmaceutical formulation) will be destabilized.

## 2. Results

Functions of the spin–lattice relaxation time (Appendix C) on the total concentration of Na-7-oxodeoxycholate (*T*_1_ = *f*(*C_T_*)) for the hydrogen atoms from the three methyl groups (C18, C19, and C21) of the examined bile salt (Figure 3) contains two steps (i.e., a sudden decrease in the spin–lattice relaxation times), which means that two critical micellar concentrations can be defined in the area of the studied concentrations (Figure 4).

The most pronounced sudden decrease in the spin–lattice relaxation times (*T*_1_) is for the angular methyl groups of the steroid skeleton of the studied bile acid (somewhat more remarkable for the hydrogen atoms of the C19 methyl group at CMC1). Namely, based on the Small–Kawamura model of the primary micelle [1,31], the convex surfaces of the steroid skeletons of the micellar building units face each other. Therefore, when the monomeric form of the bile acid anion transfers to the micellar form, the local fluctuating magnetic field modifications are the most enunciated of the angular methyl groups. Hydrogens from the angular methyl groups of the steroid skeleton in their environment (primary micelle) acquire new spins (from the second micellar building unit), i.e., magnetic dipoles, which change the magnetic dipole coupling at each hydrogen atom from the angular methyl groups. In the vicinity of CMC_1_, the slightest sudden change in *T*_1_ values is for hydrogens from the C21 methyl group. As this methyl group is from the side chain of the steroid skeleton, its environment changes to a lesser extent in the micellar state when compared to the monomer state (Figure 4). There is a sudden decrease in the *T*_1_ value in the vicinity of CMC_2_ (formation of secondary micelles), and in the case of H atoms from the C21 methyl group, the sharp decrease in *T*_1_ value (Δ = 15 ms) is of a similar magnitude. This is because, in the case of H atoms from the angular methyl groups (C18 and C19: Δ =18 ms), there was a restructuring of the micelle, especially around the C21 methyl groups. The first critical micellar concentration for Na-7-oxodeoxycholate is in the interval CMC_1_ = 40–45 mM in aqueous (D_2_O) solution at 293.1 ± 0.2 K (the literature value is 42 mM [45]), while the second critical micellar concentration is in the interval CMC_2_ = 75–80 mM in aqueous (D_2_O) solution at 293.1 ± 0.2 K.

In an aqueous solution (D_2_O) of NaCl (50 mM), both critical micellar concentrations decrease with the anion of 7-oxodeoxycholic acid compared to the CMC values for self-association from an aqueous solution without added NaCl (CMC_1_ = 30–35 mM and CMC_2_ = 55–60 mM, Figure 5). The applied concentration of NaCl is below the concentration in which dehydration of OH groups is significant (200–300 mM NaCl) [1]. However, the decrease in the second critical micellar concentration in the presence of 50 mM NaCl from the aqueous (D_2_O) solution of the investigated bile salt means that in the thermodynamic process of the formation of secondary micelles, in addition to H-bonds, the hydrophobic effect can also play a role.

Due to the low scattering intensity arising from the intermicellar interactions, the dynamic light scattering (DLS) measurements could not be carried out for 7-oxodeoxycholate in the 50–100 mM concentration range. Similar results were obtained by Akram et al. examining the sodium salts of cholic and deoxycholic acids [60]. However, by applying DSL to the sodium salt of tauroursdeoxycholic acid at concentrations above 5 × CMC (800 mM NaCl in water at 293.1 ± 0.2 K), Schurtenberger et al. managed to determine the aggregation number of the micelle. The value of the average aggregation number was ≈100 [61]. With the tested 7-oxodeoxycholate, the concentration of 5 × CMC (i.e., ≈200 mM) cannot be reached as the solution becomes cloudy.

The quenching of fluorescence of the probe (pyrene) using a quencher (hexadecylpyridinium chloride) has been explored to determine the micellar aggregation number for 7-oxodeoxycholate. In the examined bile salt and on the hydrophobic surface on the β-side of the steroid skeleton, the anion of cholic acid is reduced due to the change in the orientation of the oxygen of the oxo group of the C7 carbon about the α-axially oriented OH groups (Appendix A). Therefore, its aggregation number has a relatively small value (Figure 6). Above CMC_2_, the aggregation number increases rapidly but does not double the value of the aggregation number, which corresponds to the aggregation number between CMC_1_ and CMC_2_ (Figure 3). The presence of NaCl (50 mM) in the aqueous solution of 7-oxodeoxycholic acid anions increases the aggregation number by ≈30% in the concentration’s interval between CMC_1_ and CMC_2_, while at concentrations above CMC_2_, the aggregation number increases by ≈14%.

## 3. Discussion

In our earlier investigations, it was shown that the function *T*_1_ = *f*(*C_T_*) for bile acid anions, in addition to the C3 OH group and the C7 axial or equatorial OH (or oxo) group, does not have another OH group in a different position of the steroid skeleton because the second inflection point is missing (i.e., a sudden decrease in the spin–lattice relaxation times) [45,62]. The hydroxyl or oxo group at the C7 position of the steroid skeleton in the molecular steroid graph with the D ring is in a relative cis position. Similarly, the C12 OH group is in a relative cis position with the C17 side chain (the geometrical element, relative to which the cis or trans relationship between the structural elements of the steroid skeleton is determined, is the second-order symmetry axis of the largest longitudinal subgraph of the steroid skeleton graph [4,11], Figure 7).

Molecular fragments of the steroid skeleton in the cis position with the OH or oxo group partially screen the space around these oxygen atoms. However, as the C17 side chain is conformationally flexible, the C17 side chain can move upon the approach of another molecule (i.e., the bile acid anion as a monomeric or micellar form) during the formation of a hydrogen bond with the C12 α-axial OH group (7-oxodeoxycholic acid). In contrast, since the D ring is a constitutive part of the steroid ring system, the D ring is conformationally rigid and permanently sterically (spatially) shields the substituents from the C7 carbon of the steroid skeleton.

Conformational analysis using the Newman projection formula [63] shows that the C7 α-axial OH group is synclinal (*sk*) and the C7 oxo group is sinperiplanar (*sp*) with the D ring (Figure 8). It follows that there are steric van der Waals repulsive interactions between the synclinal and sinperiplanar groups. During hydrogen bond formation, when the OH group of the second monomer of the bile acid anion approaches the C7 α-axial OH (or oxo) group of the first monomer (anion of 7-oxodeoxycholic acid), the spatial accumulation around the C7 carbon increases (at the same time, the steric repulsion also increases). Therefore, at the anion of 7-oxodeoxycholic acid, the C3 and C12 axial OH groups and the carboxylate group from the C17 side chain of the steroid skeleton are suitable for building a hydrogen bond between two primary micelles.

According to Small’s concept, during the formation of secondary micelles of bile acid anions (${PM}_{n}\cdots {PM}_{n}={SM}_{2n}$), two primary micelles (${PM}_{n}$; equal in aggregation number, *n*) are connected by hydrogen bonds [1,28]:(2)$${PM}_{n}+{PM}_{n}↔{PM}_{n}\cdots {PM}_{n}={SM}_{2n}.$$

However, if a direct reaction of the formation of secondary micelles (2) were to occur from the primary micelles, then, even at the first critical micellar concentration, the secondary micelles would be formed in addition to the primary micelles (the law of mass action: i.e., the formation of secondary micelles would proceed parallel to the formation of primary micelles). In this case, on the function *T*_1_ = *f*(*C_T_*) of the anion of 7-oxodeoxycholic acid, two sharp decreases in the spin–lattice relaxation time would merge into one extended change in *T*_1_ (Figure 9).

In the case of primary micelles of bile acid anions, if the aggregation number is greater than three, for steric (conformational) reasons, the association of two primary micelles by hydrogen bonds between the concave surfaces of steroid skeletons is complex. Two hydrogen bonds can be formed relatively easily; the first H-bond is between the C3 α-pseudoaxial OH group of the micellar building unit from the first primary micelle and the C12 α-axial OH group of the micellar building unit from the second primary micelle. The second H-bond is between the C12 α-axial OH group of the micellar building unit from the first primary micelle and the C3 α-pseudoaxial OH group of the micellar building unit from the second primary micelle (MG, Figure 10).

However, according to Thomas and Christian, the formation of secondary micelles of bile acid anions also requires the participation of the carboxylate groups [37]. For a carboxylate group from one primary micelle to form a hydrogen bond with an α-axial OH group of another micelle (in addition to the H-bonds between the α-axial OH groups of two primary micelles), the internal hydrophobicity of one primary micelle must be broken, due to the coordinated movement of the micellar building elements from the two primary micelles and for the formation of adequate spatial angles for the creation of an H-bond (Figure 10). This leads to the hydration of the hydrophobic surfaces of the micellar building units in the steroid skeleton A ring region, which is not an entropically favorable process.

Therefore, the process of formation of the bile acid salt’s secondary micelles probably proceeds by the gradual association of monomers (m), i.e., bile acid anions, and primary micelles (${PM}_{n}$):(3)$${PM}_{n}+m↔{PMm}_{n+1},\;{PMm}_{n+1}+m↔{PMm}_{n+2},\;{PMm}_{n+2}+m↔{PMm}_{n+3},\;\ldots \;{PMm}_{n+i}+m↔{PMm}_{n+i+1}.$$

The sum of association reactions (3) is as follows:(4)$${PM}_{n}+xm↔{PMm}_{n+x},$$
where ${PMm}_{n+x}$ represents an aggregate formed by the successive association of *x* monomers (m) with the primary micelle ${PM}_{n}$ of aggregation number *n* (Figure 11).

Thermodynamic processes (3) are not entropically favorable since the effect of hydrophobic hydration of the bile acid anion occurs [55], which is bound to the primary micelle by H-bonds (Figure 11). Therefore, we propose that the degree of progression of the association reaction (4), i.e., the extent of the chemical reaction (*ξ*), has a relatively small value in the equilibrium state (*ξ* = *ξ_e_*, Figure 12). The self-association processes that comprise the resulting (collective) association reaction (3) can occur in parallel with the formation of primary micelles. However, due to the proposed relatively small value of *ξ_e_*, if the association process (3) is not coupled with an association process that has a sizeable thermodynamic driving force (i.e., de Donder’s affinity (A), Appendix D), then the equilibrium constant of the association process (3) is less than unity, and in the function *T*_1_ = *f*(*C_T_*) of the anion of 7-oxodeoxycholic acid, no second step is generated, i.e., the second critical micellar concentration is missing (Figure 4 and Figure 5). If the association processes (3) are coupled with processes that have high de Donder’s affinity, then, from the unfavorable chemical balances of the association process (3), aggregates of the type ${PMm}_{n+i+1}$ will be extracted and participate in the final process of formation of secondary micelles so that there is a continuous flow of bile acid anions (i.e., monomers) through primary micelles and ${PMm}_{n+i+1}$-type aggregates to secondary micelles.

The final process that leads to the formation of secondary micelles is the association of bile acid anions with hydrophobic surfaces (β-sides of steroid skeletons) of monomers that are hydrogen-bonded to the primary micelle, based on the existence of two separate steps on the function *T*_1_ = *f*(*C_T_*) of 7-oxodeoxycholic acid’s anion (Figure 11 and Figure 12), which must be separated from the thermodynamic processes corresponding to the first step (i.e., the first critical micellar concentration):(5)$${PMm}_{n+x}+ym↔{{PM}_{n}\cdots {PM}_{y}m}_{x-y},$$
where ${PMm}_{n+x}$ is the primary micelle with aggregation number *n* to which *x* anions of bile acids (monomer, m) were bound by H-bonds in process (3), while ${{PM}_{n}\cdots {PM}_{y}m}_{x-y}$ is an aggregate which corresponds to a general secondary micelle in which two primary micelles with different aggregation numbers are mutually bound by hydrogen bonds; *x-y* monomers remain with hydrophobic convex surfaces oriented towards the interior of the aqueous solution (Figure 13). The association of monomers (5) for the hydrophobic surface of ${PMm}_{n+x}$ aggregates explains the influence of a relatively low concentration of NaCl (50 mM in D_2_O) on the reduction in the second critical micellar concentration (Figure 5, usually higher concentrations of NaCl are needed for dehydration of OH groups [1]) and on the increase in the aggregation number (compared to the aggregation number of micelles from an aqueous solution without NaCl, Figure 6).

The process of association (5) due to the hydrophobic effect is a thermodynamically favorable process (Figure 13).

In the vicinity of the first critical micellar concentration due to the low affinity of reaction (4) (low concentration of aggregates type: ${PMm}_{n+x}$) and relatively low concentration of monomers (i.e., 7-oxodeoxycholic acid anions, which form primary micelles due to the hydrophobic effect), the association reaction (5) has a small value for the reaction extent ((*ξ_e_*)_0_), i.e., association reaction (4). A low concentration of monomers decouples the reaction for the formation of primary micelles and the association reaction (5) where secondary micelles appear (Figure 14). If the phase separation model was applied to the thermodynamic process of the formation of primary micelles of 7-oxodeoxycholate [39,64,65], then at CMC_1_ values as well as at higher total concentrations of surfactants, the monomer concentration would remain a constant value, and it would not affect the affinity of the association reaction (5). However, due to the small value of the aggregation number of the anion of 7-oxodeoxycholic acid for forming primary micelles [45] (Figure 6), the phase separation model cannot be applied. However, the law of mass action is applied [39,63,64,65,66], i.e., it is seen as a reversible chemical reaction and not the formation of a micellar pseudo-phase (Appendix E).

Affinity (a negative partial derivative of the total Gibbs free energy per the reaction extent) of the association reaction (5) at the first critical micellar concentration in the equilibrium state is as follows:(6)$${\left[{\left(\frac{\partial G}{\partial \xi }\right)}_{P,T}\right]}_{C_{m}^{0},\xi_{e}^{0}}={\left[{-A}_{P,T}\right]}_{C_{m}^{0},\xi_{e}^{0}}=0,$$
when the concentration of the monomer (i.e., the anion of 7-oxodeoxycholic acid) is $C_{m}^{0}$. With an increase in the total concentration of surfactant (7-oxo deoxycholate) according to the law of mass action for the reaction of primary micelle formation, the concentration of monomers and the concentration of primary micelles (i.e., the population of primary micelles with different aggregation numbers) simultaneously increase. According to phase separation theory, only the primary micelle concentration would increase, while the monomer concentration would remain unchanged. As the monomer concentration increases ($C_{m}^{0}+dC_{m}$), the affinity of the association reaction (5) changes. The Gibbs free energy of the system is the sum of the product of the chemical potential ($\mu_{i}$) and the amount of the corresponding particle ($n_{i}$): $G=\sum_{i}\mu_{i}n_{i}$. Therefore, the function G=f(ξ) also changes with the change in the concentration of monomers and primary micelles (Figure 14). According to Le Chatelier’s principle [67,68,69], the new equilibrium state ($\xi_{e}^{0}+d\xi$) in the association reaction is as follows:(7)$${\left[{\left(\frac{\partial G}{\partial \xi }\right)}_{P,T}\right]}_{\left(C_{m}^{0}+dC_{m}\right),\left(\xi_{e}^{0}+d\xi \right)}={\left[{-A}_{P,T}\right]}_{\left(C_{m}^{0}+dC_{m}\right),\left(\xi_{e}^{0}+d\xi \right)}=0.$$

The differential change in the affinity of the association reaction (5) between the two equilibrium states ($\xi_{e}^{0}$ and $\xi_{e}^{0}+d\xi$) is (Figure 14) as follows:(8)$$0=-dA={\left[{\left(\frac{\partial G}{\partial \xi }\right)}_{P,T}\right]}_{\left(C_{m}^{0}+dC_{m}\right),\left(\xi_{e}^{0}+d\xi \right)}-{\left[{\left(\frac{\partial G}{\partial \xi }\right)}_{P,T}\right]}_{C_{m}^{0},\xi_{e}^{0}}={\left[\frac{\partial }{\partial C_{m}}{\left(\frac{\partial G}{\partial \xi }\right)}_{P,T}\right]}_{C_{m}^{0},\xi_{e}^{0}}dC_{m}+{\left[\frac{\partial }{\partial \xi }{\left(\frac{\partial G}{\partial \xi }\right)}_{P,T}\right]}_{C_{m}^{0},\xi_{e}^{0}}d\xi .$$

According to the second-order differential rules, it is
(9)$${\left[\frac{\partial }{\partial C_{m}}{\left(\frac{\partial G}{\partial \xi }\right)}_{P,T}\right]}_{C_{m}^{0},\xi_{e}^{0}}={\left[\frac{\partial }{\partial \xi }{\left(\frac{\partial G}{\partial C_{m}}\right)}_{P,T}\right]}_{C_{m}^{0},\xi_{e}^{0}}.$$

So, from expression (8), we obtain the following:(10)$$\left[\frac{d\xi }{dC_{m}}\right]=-{\left[\frac{\partial }{\partial \xi }{\left(\frac{\partial G}{\partial C_{m}}\right)}_{P,T}\right]}_{C_{m}^{0},\xi_{e}^{0}}{\left[\frac{\partial^{2}G}{\partial \xi^{2}}\right]}_{C_{m}^{0},\xi_{e}^{0}}^{-1},$$
or with the introduction of de Donder’s chemical reaction affinity (Appendix D):(11)$$\left[\frac{d\xi }{dC_{m}}\right]={\left[\frac{\partial A}{\partial C_{m}}\right]}_{C_{m}^{0},\xi_{e}^{0}}{\left[-\frac{\partial A}{\partial \xi }\right]}_{C_{m}^{0},\xi_{e}^{0}}^{-1}.$$

During the progression of the chemical reaction, the affinity of the observed reaction decreases, which means that the last differential in Equations (10) and (11) is as follows:(12)$${\left[-\frac{\partial A}{\partial \xi }\right]}_{C_{m}^{0},\xi_{e}^{0}}^{-1}>0.$$

In light of expression (12), in order for Equations (10) and (11) to be positive ($\left(d\xi /dC_{m}\right)>0.$), i.e., in order for the equilibrium state of the association reaction (5), with an increase in monomer concentration, to move toward higher values of the reaction extent (Figure 14), the affinity differential by monomer concentration must be as follows:(13)$${\left[\frac{\partial A}{\partial C_{m}}\right]}_{C_{m}^{0},\xi_{e}^{0}}=-{\left[\frac{\partial }{\partial C_{m}}{\left(\frac{\partial G}{\partial \xi }\right)}_{P,T}\right]}_{C_{m}^{0},\xi_{e}^{0}}=-{\left[\frac{\partial }{\partial \xi }{\left(\frac{\partial G}{\partial C_{m}}\right)}_{P,T}\right]}_{C_{m}^{0},\xi_{e}^{0}}>0.$$

As the concentration of monomers increases, the association reaction (5) shifts to the right so that in the vicinity of the second critical micellar concentration, the concentration of the product of reaction (5) increases sharply, i.e., the concentration of secondary micelles, which then causes a sudden change in the *T*_1_ values of the function *T*_1_ = *f*(*C_T_*). The affinity for the association reaction (5) is (Appendix D) as follows:(14)$$A={y\mu }_{m}+\mu_{{PMm}_{n+x}}-\mu_{{{PM}_{n}\cdots {PM}_{y}m}_{x-y}}.$$

With an increase in the total concentration of 7-oxodeoxycholic acid anions (i.e., in general, bile acid anions = surfactant), according to the law of mass action (Appendix E), the equilibrium concentration of the monomer form of the surfactant increases, as well as the concentration of the primary micelle and ${PMm}_{n+x}$-type aggregates (association reaction (4)). With the increase in equilibrium concentrations of monomers and ${PMm}_{n+x}$ aggregates (Figure 11), the total hydrophobic surface in the aqueous solution increases (the sum of hydrophobic surfaces from the convex sides of the steroid skeletons of bile acid anions from the solution and ${PMm}_{n+x}$ aggregates), i.e., the hydration of the convex hydrophobic surfaces of steroid skeletons occurs, which destabilizes the system (aqueous solution of monomers and micelles). Therefore, the Gibbs free energy of the system increases (with the destabilization of the system) with an increase in the number of monomers (in the solution) as well as with an increase in the amount of ${PMm}_{n+x}$ aggregates, i.e., their chemical potential increases ($\mu_{m}$ and $\mu_{{PMm}_{n+x}}$):(15)$$\frac{{\partial \mu }_{m}}{\partial C_{m}}>0,$$
(16)$$\frac{{\partial \mu }_{{PMm}_{n+x}}}{\partial C_{m}}>0;\;\frac{{\partial \mu }_{{PMm}_{n+x}}}{\partial C_{{PMm}_{n+x}}}>0.$$

The increase in the amount of secondary micelle (${{PM}_{n}\cdots {PM}_{y}m}_{x-y}$) due to the hydrophobic effect stabilizes the system (aqueous solution), i.e., the Gibbs free energy of the system decreases:(17)$$\frac{{\partial \mu }_{{{PM}_{n}\cdots {PM}_{y}m}_{x-y}}}{\partial C_{m}}<0;\;\frac{{\partial \mu }_{{{PM}_{n}\cdots {PM}_{y}m}_{x-y}}}{\partial C_{{{PM}_{n}\cdots {PM}_{y}m}_{x-y}}}<0.$$

According to Equation (14), the effects mentioned above provide a positive value for the affinity of the association reaction (5), i.e., the positive value of the partial differential of affinity by monomer concentration (13). Therefore, due to the formation of relatively small primary micelles of 7-oxodeoxycholic acid anion [45], the law of mass action is applied for the self-association of primary micelles (i.e., the formation of primary micelles is viewed as an association reaction), according to which the equilibrium concentration of monomers increases with the increase in the total concentration of surfactants which, according to Le Chatelier’s principle, moves the association reaction (5), i.e., reaction of the formation of secondary micelles to the right (towards a more significant reaction extent). The coupling of association reactions (5) and (4) has the effect that the positive affinity value of reaction (5) shifts the equilibrium of the association reaction of ${PMm}_{n+x}$-type aggregate formation also to the right.

If the phase separation model were to be used for the formation of primary micelles, then the concentration of monomers would remain a constant value (equal to the critical micellar concentration) during the increase in the total concentration of surfactants, so the following would apply:(18)$${\left[\frac{\partial A}{\partial C_{m}}\right]}_{C_{m}^{0},\xi_{e}^{0}}=0\;\to \;\left[\frac{d\xi }{dC_{m}}\right]=0.$$

## 4. Materials and Methods

3α,12α-Dihydroxy-7-oxo-5β-cholanoic acid (7-oxodeoxycholic acid) was obtained according to Tullar from the cholic acid (Sigma, Auckland, New Zealand; purity ≥ 99%) [70]. The 7-oxodeoxycholic acid was transformed into sodium salts by a known procedure [6]. Sodium cholate (≥98%, Sigma Aldrich, St. Louis, MO, USA), pyrene (≥98%, Sigma-Aldrich, Munich, Germany), hexadecylpyridinium chloride (98%, Merck, Darmstadt, Germany) and NaCl (≥98%, Sigma-Aldrich, Munich, Germany) were used as received.

### 4.1. ^1^H NMR Studies

The concentrations of the examined bile salt in D_2_O were in the interval of 5–100 mM (in steps of 5 mM), while the concentrations of the examined bile salt in D_2_O + 50 mM NaCl were in the interval of 5–90 mM, also in steps of 5 mM. Solutions were obtained by diluting a stock solution of 100 mM Na-7-oxodeoxycholate in D_2_O (or 90 mM Na-7-oxodeoxycholate in D_2_O + 50 mM NaCl) with D_2_O or a solution of D_2_O + 50 mM NaCl. Measurements were performed at 293.1 ± 0.2 K on a Bruker Spectrospin-500 (Rheinstetten, Germany) instrument with standard Bruker software. The ^1^H NMR spectra were recorded using a spectral window of 3200 Hz. Spin–lattice relaxation times *T*_1_ were determined by the inversion recovery experiments (180°-*τ*-90°-AQC) [45,62,71]. Selected peak areas for nine different interpulse delays *τ* were determined. Each spectrum was recorded after a certain time *τ* upon applying a pulse of 180°: the first spectrum was recorded after 0.010 s, the second after 0.025 s, and so on (Appendix F). The relaxation time was determined by fitting experimental data to the following equation (*I*—signal surface; the relative standard uncertainty for the *τ* is 5%):(19)$$I=I_{0}\left(1-2\exp\left[\frac{-\tau }{T_{1}}\right]\right).$$

### 4.2. Fluorescence Measurements—Determination of the Micelle Aggregation Numbers

The micelle aggregation numbers were determined using a steady-state fluorescence quenching technique using a Shimadzu spectrofluorimeter-5000 (Tokyo, Japan) with excitation and emission slit widths of 3 nm. Pyrene and hexadecylpyridinium chloride were used as probes and quenchers. In the Na-7-oxodeoxycholate solution (water or NaCl aqueous solution (50 mM)), the pyrene concentration was kept constant at 3 × 10^−6^ M [60]. The excitation wavelength was kept at 337 nm, and emission spectra were recorded in the 350–450 nm range. Measurements were performed at 293.1 ± 0.2 K temperature. The relative standard uncertainty of the aggregation number in the concentration interval 50–80 mM is 7% and in the interval 90–100 mM is 10%.

### 4.3. Dynamic Light Scattering (DLS) Measurements

Dynamic light scattering measurements were performed using a laser-spectroscatter 201 from RiNA GmbH (Berlin, Germany). In DLS measurements, a laser beam was guided toward the sample under investigation, with a fixed detection arrangement of 90° to the center of the cell area, and the fluctuation in the intensity of the scattered light was measured.

## 5. Conclusions

For the anion of 7-oxodeoxycholic acid, the function *T*_1_ = *f*(*C_T_*) has two sudden changes in the *T*_1_ value (293.1 ± 0.2 K). The first corresponds to the critical micellar concentration (CMC_1_) at which primary micelles are formed, while the second corresponds to the critical micellar concentration (CMC_2_) where secondary micelles are formed. The aggregation number of the micelle changes abruptly in the concentration interval of 80–90 mM.

For the existence of two separate steps (jumps) on the function *T*_1_ = *f*(*C_T_*), direct association of primary micelles with H-bonds is thermodynamically not possible (as well as for stereochemical reasons). The formation of secondary micelles takes place via two main processes. Process I corresponds to the association of monomers through H-bonds for primary micelles—a thermodynamically unfavorable process (${PMm}_{n+x}$, with the newly associated monomer, the convex hydrophobic surface is exposed to hydration); in process II, the aggregate from process I, through the hydrophobic effect, binds a new monomer, thus forming a secondary micelle—a thermodynamically favorable process. At relatively low concentrations of monomers (around CMC_1_), the equilibrium concentration of aggregate ${PMm}_{n+x}$ is low, which then decouples the processes of formation of primary and secondary micelles. As the anions of 7-oxodeoxycholic acid form micelles with a relatively small aggregation number, the law of mass action applies to their association. So, with the increase in the total concentration of surfactants, the concentration of monomers also increases. According to Le Chatelier’s principle, this shifts the balance of association (5) to the right, and, therefore, aggregates of the type ${PMm}_{n+x}$ are extracted from process of association (3)—coupling of the formation of primary and secondary micelles due to the increase in monomer concentration.

## Figures and Tables

**Figure 1 ijms-24-11853-f001:**
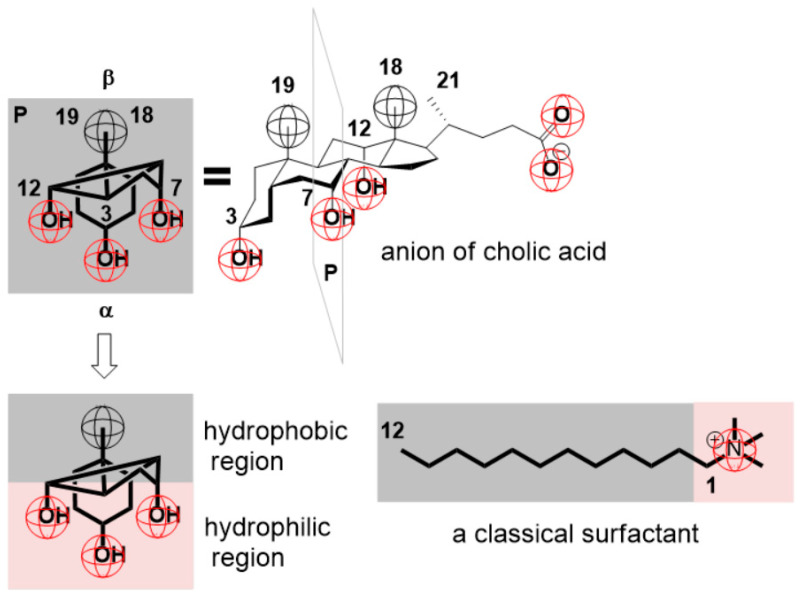
The conformation of cholic acid anion and the projection of the steroid skeleton atoms into the P plane, the axial OH groups on the concave surface of the steroid skeleton (α-side) form a hydrophilic region—planar polarity.

**Figure 2 ijms-24-11853-f002:**
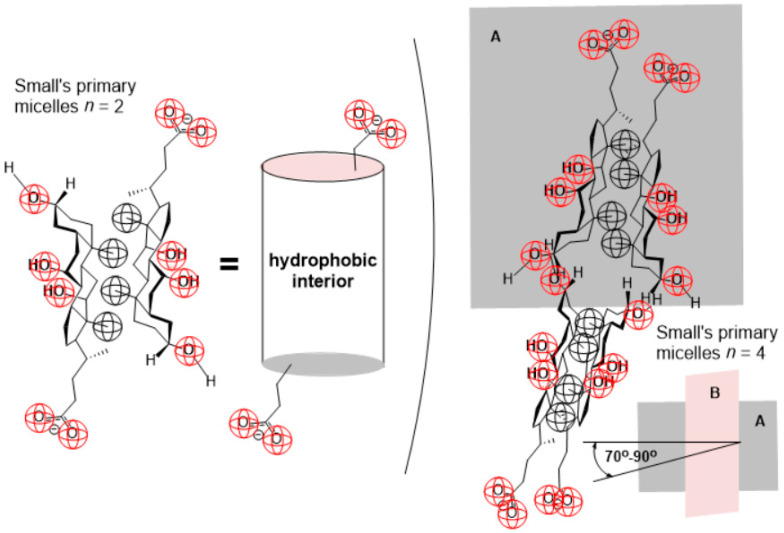
Small’s primary micelles: dimeric micelle—carboxylate groups are on opposite sides of the aggregate; tetrameric micelles are formed by the back-to-back association of two monomers, where the carboxylate groups are on the same side of the aggregate, two such aggregates are connected via regions of the A ring of the steroid skeleton so that planes A and B (planes that are normal to the middle plane of the steroid skeleton) overlap a certain angle, two pairs of carboxylate groups are in the trans position.

**Figure 3 ijms-24-11853-f003:**
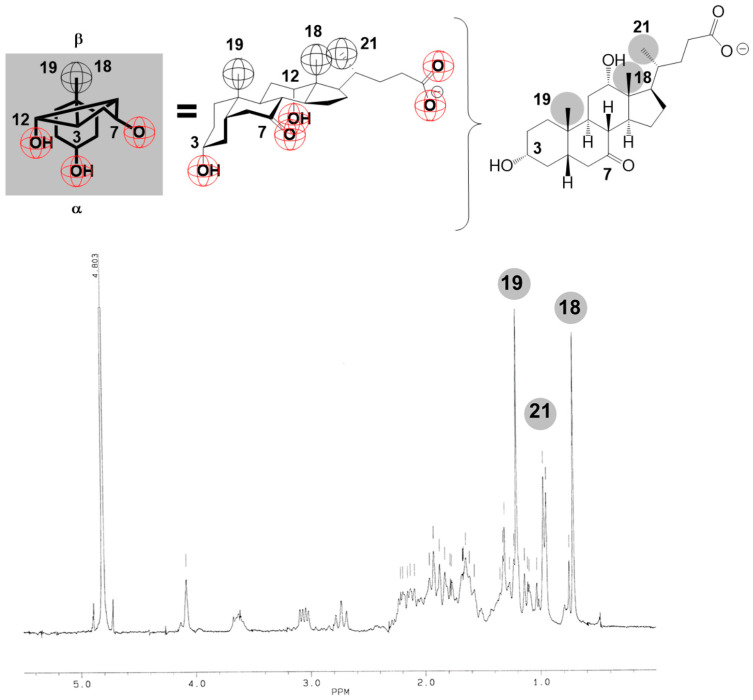
The molecular and conformational formula of the anion of 7-oxodeoxycholic acid and the ^1^H NMR spectrum of Na-7-oxodeoxycholate in D_2_O (35 mM), the methyl groups whose H atoms are determined by the spin–relaxation times are marked (293.1 ± 0.2 K).

**Figure 4 ijms-24-11853-f004:**
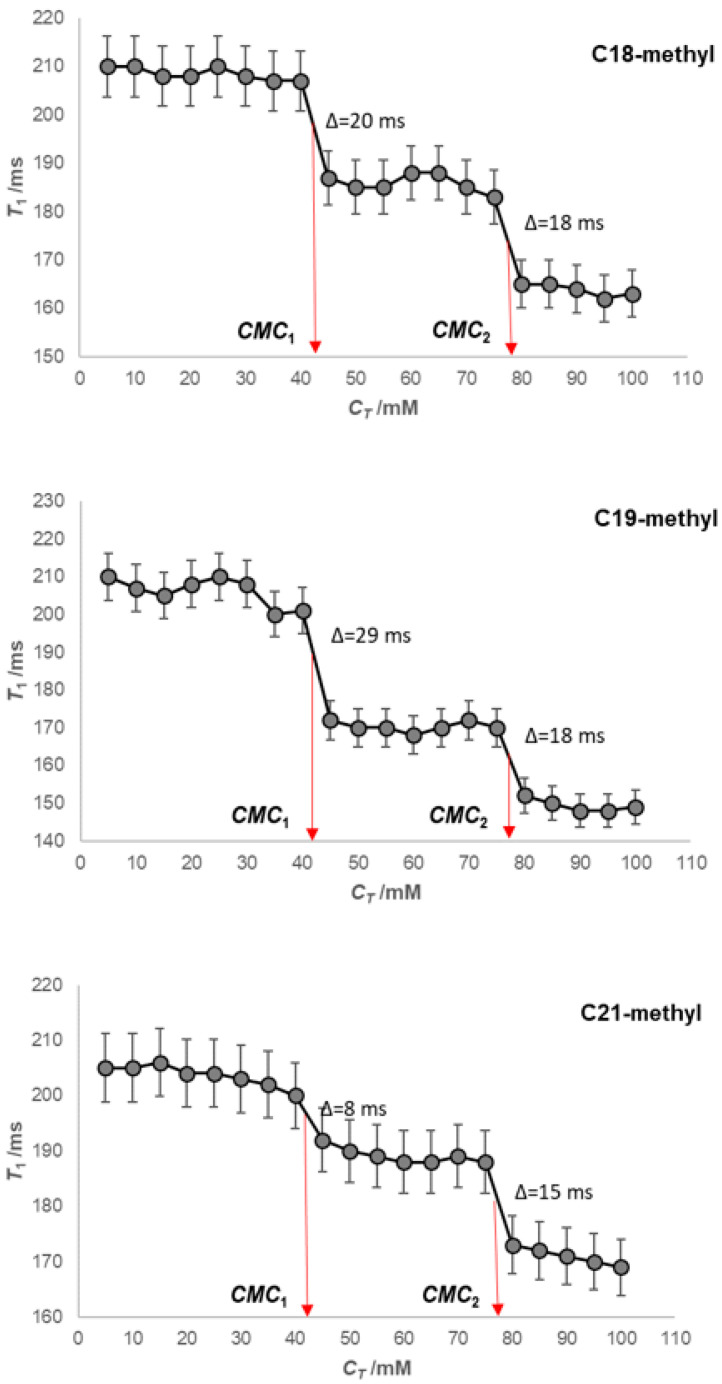
Dependences of spin–lattice relaxation times (*T*_1_) on the total concentration of Na-7-oxodeoxycholate for hydrogen atoms with C18, C19, and C21 methyl groups at 293.1 ± 0.2 K (aqueous solution (D_2_O) without NaCl).

**Figure 5 ijms-24-11853-f005:**
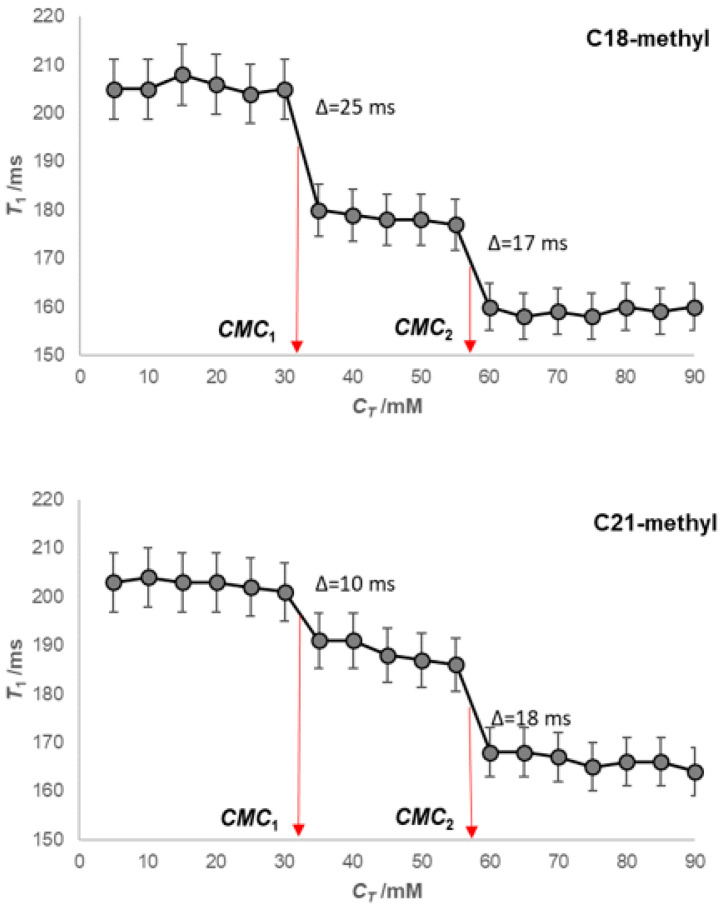
Dependences of spin–lattice relaxation times (*T*_1_) on the total concentration of Na-7-oxodeoxycholate for hydrogen atoms with C18 and C21 methyl groups in NaCl aqueous (D_2_O) solution (50 mM) at 293.1 ± 0.2 K.

**Figure 6 ijms-24-11853-f006:**
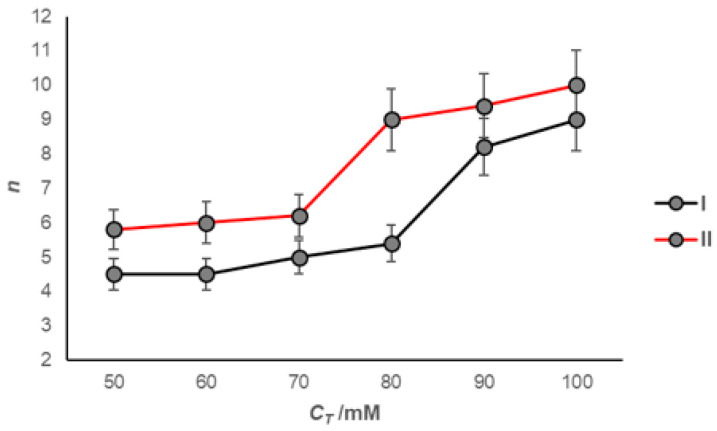
Dependence of the aggregation number (*n*) of the micelle on the total concentration of Na-7-oxodeoxycholate; I in water and II in NaCl aqueous solution (50 mM) at 293.1 ± 0.2 K.

**Figure 7 ijms-24-11853-f007:**
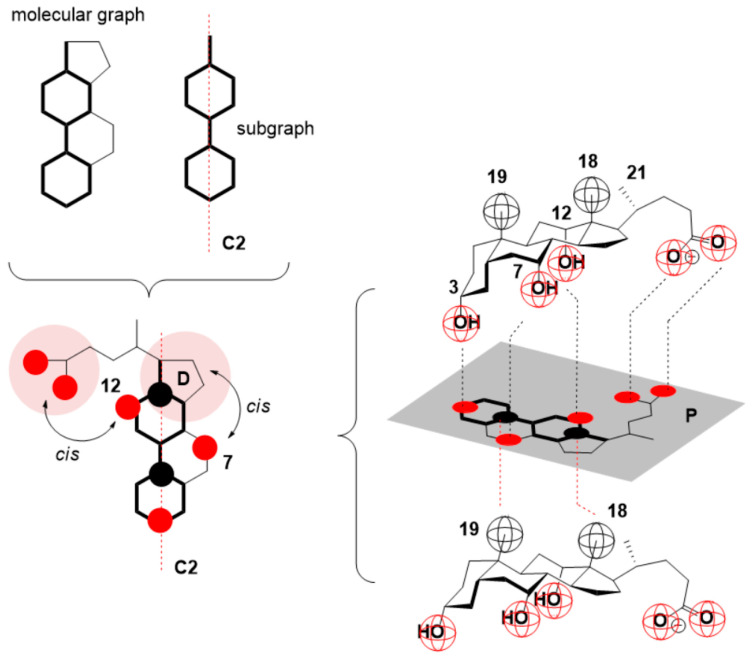
In the molecular graph of the steroid skeleton, the C7 OH group is in the cis position with the conformationally rigid D ring, while the C12 OH group is also in the cis position with the conformationally flexible C17 side chain; the second-order symmetry axis of the largest longitudinal subgraph of the steroid skeleton is a geometric element for determining the relative cis–trans relationship (P = plane of projection).

**Figure 8 ijms-24-11853-f008:**
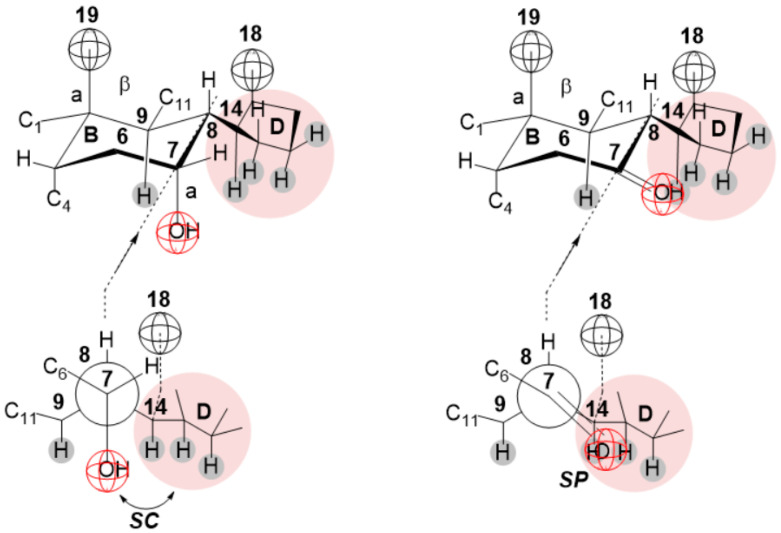
Conformational analysis of the environment of C7 α-axial and C7 oxo groups using Newman’s projection formulas (*sc*—synclinal and *sp*—sinperiplanar).

**Figure 9 ijms-24-11853-f009:**
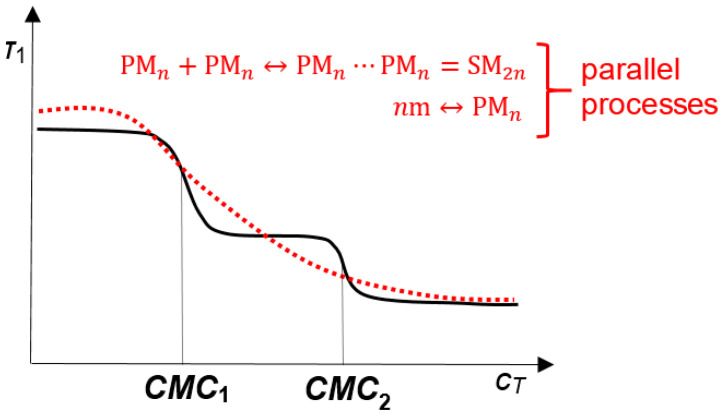
The dependence *T*_1_ = *f*(*C_T_*), obtained experimentally (solid line) and based on the law of mass action calculations (dashed line).

**Figure 10 ijms-24-11853-f010:**
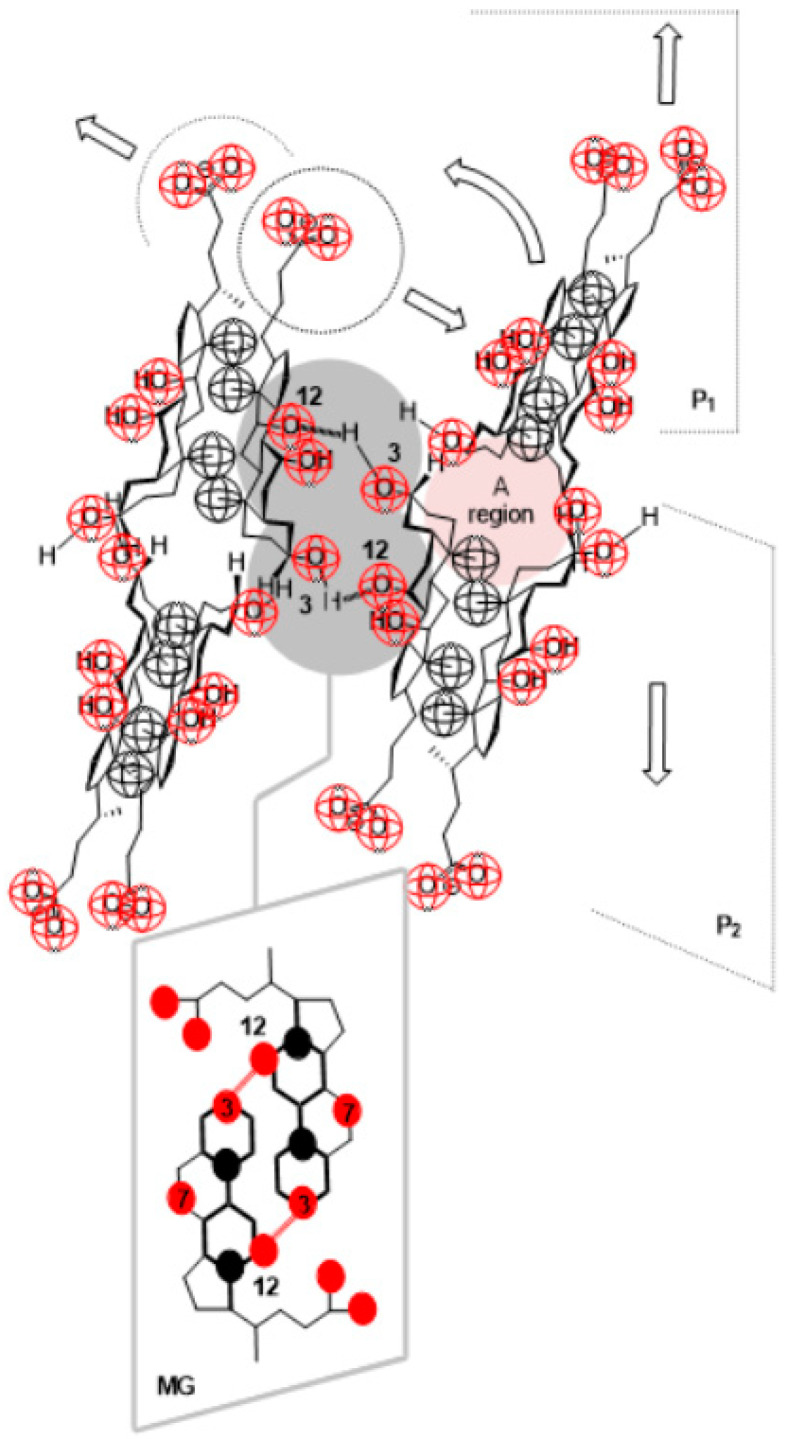
Between two primary micelles, the formation of a secondary micelle is complex.

**Figure 11 ijms-24-11853-f011:**
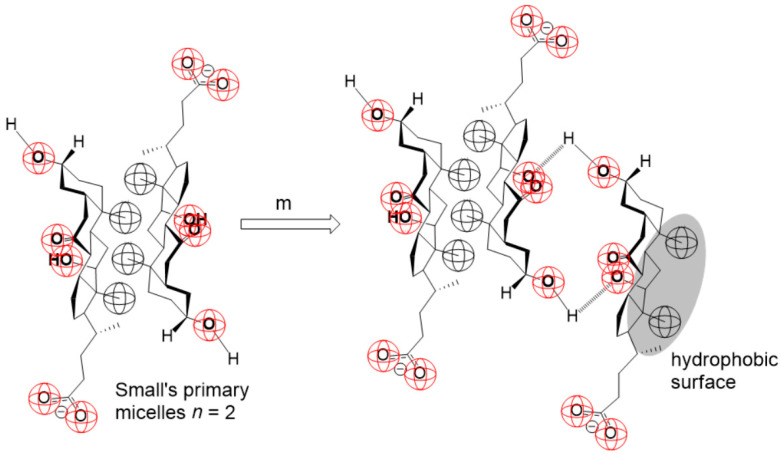
An example of the association of monomer (m = anion of 7-oxodeoxycholic acid) using H-bonds for Small’s dimeric primary micelle (${PM}_{2}+m↔{PMm}_{2+1}$); in the resulting aggregate, the micellar building unit’s hydrophobic surface on the β-side of the steroid skeleton is exposed to hydration, i.e., it is not shielded with other building units—aggregate destabilization due to hydrophobic hydration.

**Figure 12 ijms-24-11853-f012:**
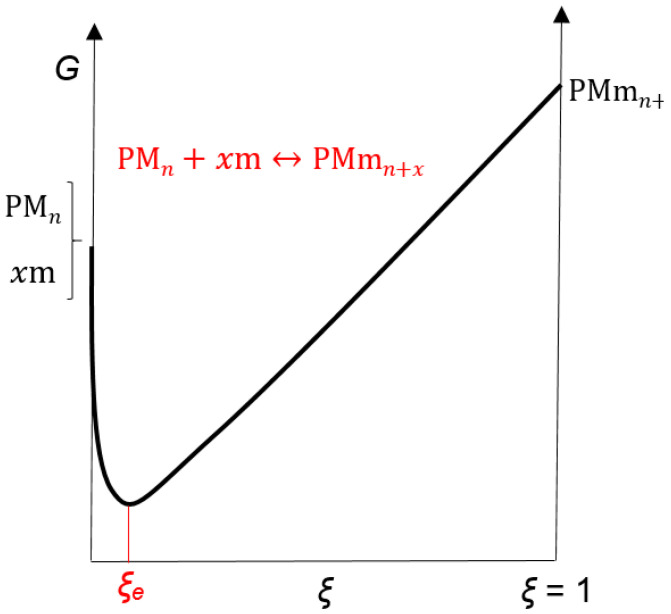
The monomer association process (via H-bond) for the primary micelle—a thermodynamically unfavorable process.

**Figure 13 ijms-24-11853-f013:**
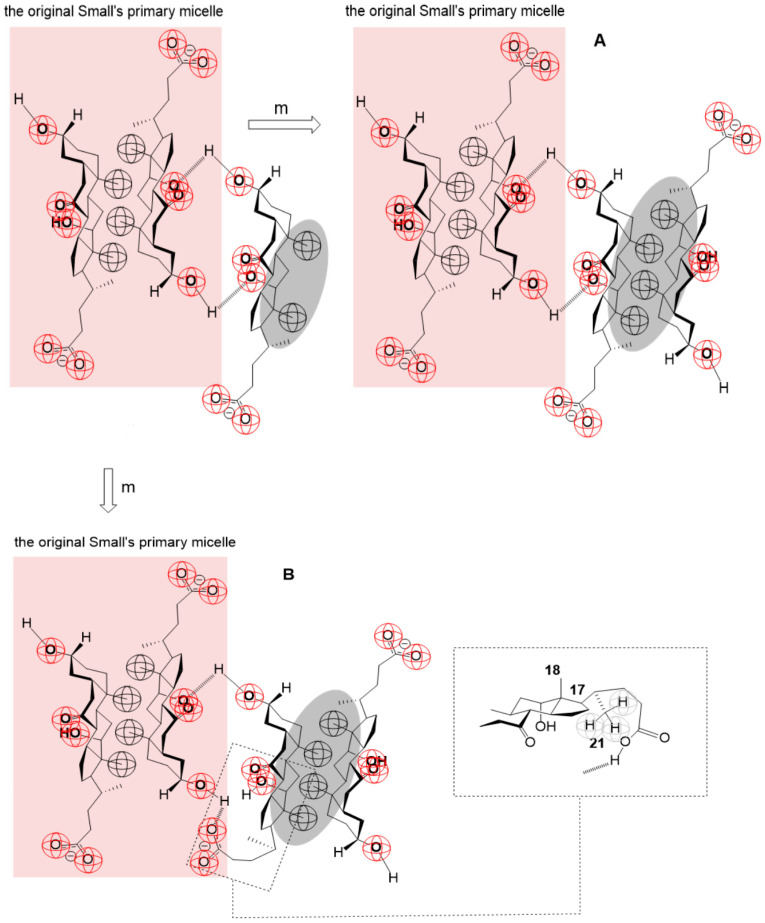
The simplest example for the process of forming a secondary micelle (anion of 7-oxodeoxycholic acid) by binding monomers to the hydrophobic surface of ${PMm}_{n+i+1}$ aggregates: hydrophobic effect: (**A**) between the primary micelles there are H-bonds that form the OH groups of the steroid skeleton; (**B**) between the primary micelles there are H-bonds formed by carboxylate and OH groups, the C21 methyl group changes its environment if the carboxylate group C17 of the side chain participates in the formation of a hydrogen bond.

**Figure 14 ijms-24-11853-f014:**
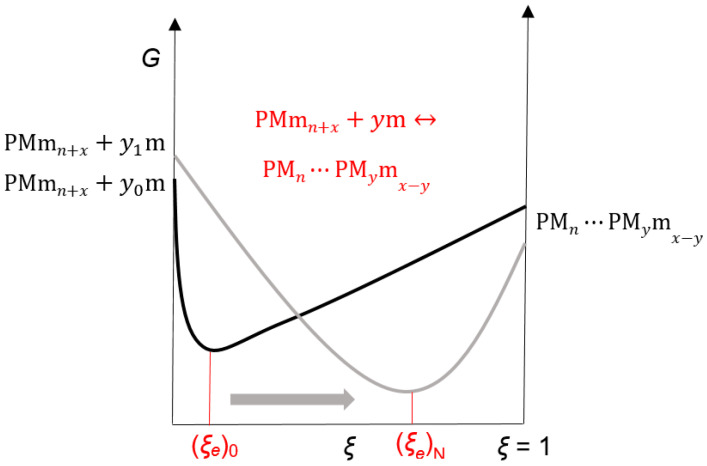
With an increase in the total concentration of surfactants, according to the law of mass action, the concentration of monomers also increases, which results in a new equilibrium state in the association reaction (5).

## Data Availability

Not applicable.

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
