# Peer review of "Self-Association of the Anion of 7-Oxodeoxycholic Acid (Bile Salt): How Secondary Micelles Are Formed"

_ijms, 2023, doi:10.3390/ijms241411853_

Round 1

Reviewer 1 Report

This is an interesting study that delineates the self-assembly behavior of charged surfactants and contributes towards understanding of the overall process leading to defined and ordered aggregates. Theoretical evaluation is very well performed, and the discussion goes through detailed analyses of different models. I would recommend its publication with revisions.

The introduction is too long, and one gets lost in the process as to what is the author really getting at. It distracts the reader and will eventually take them away from an interesting story. I would strongly suggest that the author trims this part, clearly define the mechanistic description of different stages that can lead to primary and secondary micelles, what are the challenges in determining this trajectory, and subsequently bring the goals of this study into focus.

Discussion is the main important part of this study and suggests a dynamic process with two different pathways leading to secondary micelle structures. How could an experimentalist really prove this? Critical micelle concentrations depend on a number of factors and we can simplify to the extent as described by the author, there is no problem there. But could this be confirmed and additionally supported by techniques such as NMR/IR/DLS?

There are several typographical errors in the manuscript which the author should consider removing to keep the reader on track and interested. Some examples are noted below:

In both steroisomeric cholanic acid…..

…. such as deoxycholic, lithocholic, and hyodeoxycholic acids (Figure 4)….

Accordinc Small the bile acids…..

…..the dimeric Small's micelle plays a crucial role….

Functions of dependence of the spn-lattice relaxation time (Appendix A)…..

And so on….

Small proposed the concept of the for-…… (Give a reference to this statement)

Suggestions have been made in the original report.

Author Response

The authors are very thankful for the Referee's comments and suggestions.

  1. The introductory part has been shortened, and some Figures and explanations related to the stereochemistry of the steroid skeleton, which is important for the hydrophobicity and self-association of bile salts, have been moved to the Appendix section.
  2. Due to the small size of the 7-oxodeoxycholic acid micelle, the DSL experiment did not give further results. Therefore, the aggregation number of the micelle was determined by the quenching of fluorescence of the probe (pyrene) by a quencher (hexadecylpyridinium chloride), (line: 224).
  3. The English language was corrected using the MDPI editing service.

Reviewer 2 Report

Self-association of the anion of 7-oxodeoxycholic acid (bile salt): how the secondary micelles are formed

Mihalj Poša

 Based on a review of the literature data and their own experimental results obtained by 1H NMR spectroscopy, authors analyzed the self-association of anions of bile acids. Modern theoretical concepts are used to explain the successive concentration transition of primary micelles (in the region of critical micelle concentration, CMC1) into larger secondary micelles (in the region of CMC2). The work is of interest for researchers in the field of supramolecular chemistry and also allows a deeper understanding of the specifics of aggregation processes in biological systems.

However, there are questions and comments to the work:

1. Abstract

lines 10-11:

Phrase «Knowing the bile salts aggregates that form during self-association for pharmaceutical formulations is necessary» is incorrect (and grammatically mistaken). This article is not just about surfactant aggregates, but assumes their different structures (at different concentrations of amphiphile), as well as considers the ways of their formation. The phrase needs to be changed to reflect these features of the material.

2. Keywords.

Object of investigation is recommended to be included, e.g., bile acids.

3. Introduction

1. Urgency, novelty and practical significance of the work should be emphasized.

2. The Introduction is too long, it should be shortened, with a part of Figures transferred to the Supplementary section.

4. Results

To analyze the process of aggregation of bile acid anions, the authors use 1H NMR spectroscopy data, namely, they analyze the dependence of the spin-lattice relaxation time on the total surfactant concentration. At the same time, the question remains unanswered, how the size of aggregates changes when CMC1 and CMC2 are found by this method. The use of DLS technique is highly recommended to obtain the particle size distribution at various surfactant concentrations. This would allow to achieve a more complete information on the aggregation in the system.

5.Discussion

Some figure captions are incorrect:

A) Caption for fig. 15 should be more specific rather than repeating the text above.

For example: The dependence T1 = f(CT), obtained experimentally (solid line) and that based on the law of mass action calculations (dashed line).

B) Caption to fig. 16 needs to be more specific, and the reasoning should be moved to the text of the article.

6. Experimental.

1. The spectra given in "Materials and Methods" are recommended to be transferred to supplementary materials.

2. Please explain, what is the reason for using a high concentration of NaCl of 50mM?

Minor comments

1.     Line 518 100 mN is it correct?

2.     Line 71 «Accordinc Small the bile acids should be replaced According Small the bile acids.

Manuscript should be thoroughly revised to remove grammar mistakes.

Author Response

The authors are very thankful for the Referee's comments and suggestions.

  1. The abstract was corrected according to the reviewer's instructions.
  2. The term: bile acids was added to the keywords.
  3. The introductory part has been shortened, and some Figures and explanations related to the stereochemistry of the steroid skeleton, which is important for the hydrophobicity and self-association of bile salts, have been moved to the Appendix section. The introductory part emphasized that the proposed thermodynamic model of the formation of secondary micelles is a new model (line 140).
  4. Due to the small size of the 7-oxodeoxycholic acid micelle, the DSL experiment did not give further results. Therefore, the aggregation number of the micelle was determined by the quenching of fluorescence of the probe (pyrene) by a quencher (hexadecylpyridinium chloride), (line: 224).
  5. The titles of the proposed Figures in the discussion have been corrected according to the reviewer's instructions.
  6. The NMR spectrum from the experimental section is transferred to the Appendix.
  7. Dehydration of the OH group of the steroid skeleton is affected by NaCl at a concentration of 300 mM (and at higher concentrations). In comparison, at a concentration of 50 mM, NaCl affects the hydrophobic interaction, therefore if this concentration of NaCl changes the value of CMC2, it means that even during the formation of secondary micelles, the hydrophobic effect also participates (in addition to H-bonds) (line: 151; line: 364).
  8. The English language was corrected using the MDPI editing service.

Round 2

Reviewer 2 Report

All my comments were taken into account. Manuscript can be recommended for publication.

Author Response

The reviewer had no suggestions.